# Stretching magnetism with an electric field in a nitride semiconductor

D. Sztenkiel[1,*], M. Foltyn[1,*], G.P. Mazur[1], R. Adhikari[2,3], K. Kosiel[4], K. Gas[1,5], M. Zgirski[1], R. Kruszka[4], R. Jakiela[1], Tian Li[1], A. Piotrowska[4], A. Bonanni[2,3], M. Sawicki[1] & T. Dietl[1,6,7]

The significant inversion symmetry breaking specific to wurtzite semiconductors, and the associated spontaneous electrical polarization, lead to outstanding features such as high density of carriers at the GaN/(Al,Ga)N interface—exploited in high-power/high-frequency electronics—and piezoelectric capabilities serving for nanodrives, sensors and energy harvesting devices. Here we show that the multifunctionality of nitride semiconductors encompasses also a magnetoelectric effect allowing to control the magnetization by an electric field. We first demonstrate that doping of GaN by Mn results in a semi-insulating material apt to sustain electric fields as high as $5\,MV\,cm^{-1}$. Having such a material we find experimentally that the inverse piezoelectric effect controls the magnitude of the single-ion magnetic anisotropy specific to $Mn^{3+}$ ions in GaN. The corresponding changes in the magnetization can be quantitatively described by a theory developed here.

[1] Institute of Physics, Polish Academy of Sciences, aleja Lotników 32/46, PL 02-668 Warszawa, Poland. [2] Institut für Halbleiter- und Festkörperphysik, Johannes Kepler University, Altenbergerstrasse 69, A-4040 Linz, Austria. [3] Linz Institute of Technology, Johannes Kepler University, Altenbergerstrasse 69, A-4040 Linz, Austria. [4] Institute of Electron Technology, aleja Lotników 32/46, PL 02-668 Warszawa, Poland. [5] Institute of Experimental Physics, University of Wrocław, pl. M. Borna 9, PL 50-204 Wrocław, Poland. [6] Institute of Theoretical Physics, University of Warsaw, ulica Pasteura 5, PL 02-093 Warszawa, Poland. [7] WPI-Advanced Institute for Materials Research, Tohoku University, 2-1-1 Katahira, Aoba-ku, 980-8577 Sendai, Japan. * These authors contributed equally to this work. Correspondence and requests for materials should be addressed to T.D. (email: dietl@ifpan.edu.pl).

I t has been known for a long time that piezoelectricity specific to crystals with no inversion symmetry offers a spectrum of outstanding functional properties. For instance, various nanoelectromechanical and energy harvesting devices employing wurtzite II–VI and III–V semiconductor compounds have recently been demonstrated[1]. At the same time, nano-composite[2,3] or hybrid[4,5] piezoelectric/magnetic systems allow for the electric control of magnetization.

In $Ga_{1-x}Mn_xN$ system a single-ion magnetic anisotropy dominates for $Mn^{3+}$ ions that assume in GaN a high-spin configuration with spin and orbital momentum $S=2$ and $L=2$, respectively[6,7]. For the experimental values of the lattice parameters $a$ and $c$ the easy axis is perpendicular to the polar [0001] direction ($c$ axis) of the wurtzite structure[6,7]. At the same time a mid-gap $Mn^{2+}/Mn^{3+}$ level traps electrons introduced by Si or residual donors, so that the material becomes semi-insulating for $x \gtrsim 0.05\%$ (refs 8,9). Because of Anderson-Mott and Hubbard-Mott localization of carriers residing in the Mn impurity band, the semi-insulating character of $Ga_{1-x}Mn_xN$ persists up to the highest available Mn concentrations. In this dilute magnetic insulator, the exchange coupling between $Mn^{3+}$ ions is dominated by short-range ferromagnetic superexchange interactions leading to a ferromagnetic ordering at $T_C \lesssim 1\,K$ for $x \lesssim 3\%$ (ref. 10).

Here we find experimentally the existence of a strong coupling between piezoelectricity and magnetism in the dilute magnetic insulator $Ga_{1-x}Mn_xN$ and develop a theory that describes the effect quantitatively. The measurements are performed at $T \geq 2\,K$ and $x \lesssim 3\%$, that is in the paramagnetic regime. Under these conditions, the inverse piezoelectric effect by stretching the elementary cell along the $c$ axis affects the magnitude of the uniaxial magnetic anisotropy and, thus, of the magnetization. We reveal this novel magnetoelectric effect by direct magnetization measurements employing a new detection method. Our work bridges two fields of research so far independent: piezoelectricity of wurtzite semiconductors and electric control of magnetization in magnetic insulators. More specifically, the findings presented in this Communication lead to the conclusions: (i) the magnetoelectric effect generated by piezoelectricity can exist in a homogeneous crystalline compound, not only in hybrid or nanocomposite systems, (ii) the multifunctional capabilities of $Ga_{1-x}Mn_xN$ extend into the core of spintronic functionalities, like the manipulation of magnetization by the electric field.

## Results

**Samples**. The observation of the piezoelectromagnetic effect (PEME) is built on our recent progress in the metal-organic vapour phase epitaxy (MOVPE) and nanocharacterization of wurtzite $Ga_{1-x}Mn_xN$ and $Ga_{1-x}Mn_xN$:Si layers, which led us to fabricate a single phase homogenous alloy with Mn atoms substituting Ga sites up to $x=3.5\%$ (ref. 8). The structures studied here are grown by MOVPE onto $c$-plane 2″ sapphire substrates and have the $c$ axis parallel to the growth direction and Ga-face polarity. They consist of $1\,\mu m$ GaN buffer, a 500 nm-thick Si doped $n^+$-GaN bottom contact layer ($n \gtrsim 10^{19}\,cm^{-3}$), and a 30 nm-thick $Ga_{1-x}Mn_xN$ or $Ga_{1-x}Mn_xN$:Si film to which a vertical electric field can be applied. Structures with three different films have been deposited, namely: $Ga_{1-x}Mn_xN$, $Ga_{1-x}Mn_xN$:Si having $x \simeq 2.4$ and 2.5% as established from magnetization data—denoted as A and B, respectively—and a reference structure containing $Ga_{1-x}Mn_xN$:Si with $x=0.1\%$ (denoted as R). Samples of dimensions $(5 \times 5)\,mm^2$ are cut out from the wafers, and sys-tematically characterized by atomic force microscopy (AFM), high-resolution x-ray diffraction (HRXRD), secondary-ions mass

spectrometry, high-resolution transmission electron microscopy (HRTEM), and superconducting quantum interference device (SQUID) magnetization measurements.

Details on the instrumentation and on the results of the comprehensive structural and chemical characterization of the samples are collected in Methods. Both HRXRD and HRTEM images point to a high crystalline quality of the wurtzite structures, as well as reveal neither secondary phases nor relaxation defects in strained $Ga_{1-x}Mn_xN$ layers. According to the high-angle annular dark-field TEM images, $Ga_{1-x}Mn_xN$ layer thickness is $t=32\,nm$ with no indication of chemical phase separation (Mn aggregation) that might be driven by spinodal decomposition. This magnitude of $t$ is consistent with secondary ion mass spectrometry (SIMS) data that corroborate also the values of Mn concentrations deduced from magnetization measurements on samples A, B and R discussed in the next subsection. The magnetization data confirm also that the Mn distribution is uncorrelated and that an overwhelming majority of Mn ions assumes the $3+$ charge state specific to the Ga-substitutional case. Finally, we note that AFM topography reveals with the root mean square surface roughness up to 8 nm and lateral extensions in the µm range. This roughness may results in a certain inhomogeneity of the electric field applied to the structure.

**Magnetization as a function of the magnetic field**. Because of a relatively small film thickness and low Mn content, an accurate determination of magnetization $M(T, H)$ for our $Ga_{1-x}Mn_xN$ layers by SQUID magnetometry has required an extension of the previous experimental methodology[11]. The modifications are outlined in the section Methods.

The circles in Fig. 1 represent the experimental data $M(T, H)$ per unit sample area $A$ for the A and B $Ga_{1-x}Mn_xN$ films at two orientations of the magnetic field $H$ with respect to the $c$ axis. No hysteretic behaviour is found down to the experimental noise level $[\sigma_{M/A}(H \simeq 0) \lesssim 1 \times 10^{-6}\,emu\,cm^{-2}$, where $A \simeq 0.25\,cm^2]$ at any investigated temperature (2, 5, 10 K—presented in Fig. 1—and at 50 and 300 K, not shown). This points to the absence of a ferromagnetic phase that might originate from Mn-rich aggregates. This conclusion is further supported by results of HRXRD and HRTEM studies on these samples presented in Methods, and which—within the ultimate limits specific to these techniques—rule out the presence of any Mn-rich phases either in the form of nanometric precipitates or of regions of spinodal decomposition[12].

Actually, both layers exhibit magnetism characterized by $M(T, 0) = 0$, $M(T, H \rightarrow \infty)$ tends to saturate, and $\chi(T) = dM(T, H \rightarrow 0)/dH$ increases with lowering temperature according to $\chi(T) \propto 1/(T - T_0)$. This behaviour demonstrates that the studied samples at $T \geq 2\,K$ are in the paramagnetic phase, though ferromagnetic coupling between neighbouring Mn spins starts to be visible and leads to $T_0 > 0$. At the same time, a sizable uniaxial magnetic anisotropy is observed, $M(T, \mathbf{H} \perp \mathbf{c}) > M(T, \mathbf{H} \parallel \mathbf{c})$, which is specific for $Mn^{3+}$ ions in wurtzite GaN (refs 6,7).

The starting point of the theory aiming in description of $M(T, H)$ in the paramagnetic phase is a single-ion Hamiltonian in the form implied by the group theory for $Mn^{3+}$ ions in wurtzite GaN (refs 6,7). This Hamiltonian and the corresponding crystal-field, spin-orbit and Jahn-Teller parameters employed to generate theoretical curves in Fig. 1 are presented in Methods. The theoretical procedure, developed earlier for purely paramagnetic $Ga_{1-x}Mn_xN$ ($x < 0.6\%$) containing $Mn^{3+}$ ions[7], is modified here to describe samples with higher Mn concentrations. In particular, we introduce an effective temperature $T' = T - T_0(T)$ as a fitting parameter, where

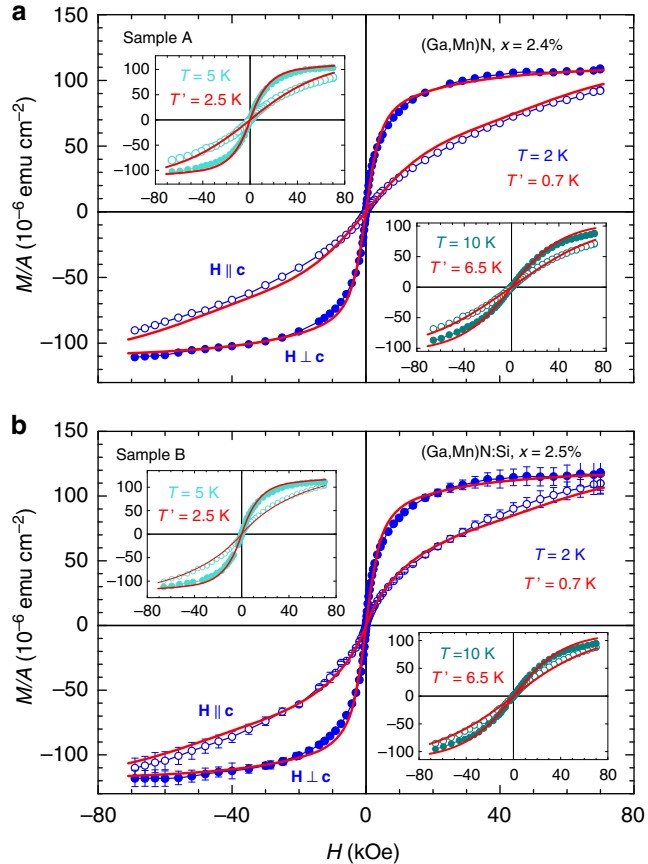

**Figure 1 | Areal density of magnetic moment.** (**a**) $Ga_{1-x}Mn_xN$ (sample A) and (**b**) $Ga_{1-x}Mn_xN$:Si (sample B). Results are shown for selected temperatures $T$ as a function of the magnetic field $H$ applied parallel (open squares) and perpendicular (closed circles) to the wurtzite $c$ axis. For clarity, experimental error bars are provided only for measurements at 2 K for sample B. Solid lines: magnetization curves calculated for $Mn^{3+}$ ions (sample A) and for $Mn^{3+}$ and $Mn^{2+}$ ions in the case of sample B. $T'$ are the values of effective temperatures adjusted to account for ferromagnetic $Mn^{3+} - Mn^{3+}$ interactions.

$T_0(T) > 0$ takes phenomenologically into account the presence of spin-spin coupling in dilute ferromagnets above $T_C$. While another fitting parameter $x$ controls the magnitude of saturation magnetization, the Crurie-Weiss-like temperature $T_0(T)$ determines, at given $x$, the magnitude of the slope $dM(T, H \to 0))/dH$.

Especially relevant for our work is the quantitative interpretation of magnetic anisotropy. The single-ion magnetic anisotropy is virtually negligible for $Mn^{2+}$ ions, as the orbital momentum vanishes in this case ($S = 5/2$, $L = 0$). In contrast, for $Mn^{3+}$ ions in GaN ($S = 2$, $L = 2$) a sizable single-ion uniaxial magnetic anisotropy has been reported[6,7]. In GaN:Mn, the uniaxial anisotropy originates primarily from the trigonal deformation of the tetrahedron formed by anion atoms next to Mn, as illustrated in the panels b and c of Fig. 2. In terms of the lattice parameters $a$ and $c$, the deformation is described by the parameter $\xi = c/a - \sqrt{8/3}$. In particular, for $\xi < 0$, the case of GaN:$Mn^{3+}$, the easy axis is perpendicular to the wurtzite $c$ axis, whereas in the opposite case $\xi > 0$, the easy axis assumes the direction of the $c$ axis. According to the theoretical model of $Mn^{3+}$ ions (see Methods), the effect of trigonal deformation is described by two crystal field parameters, $B_2^0$ and $B_4^0$ which, for the relevant small values $\xi$, can be assumed to be linear in $\xi$, $B_i^0 = y_i \xi$. The values of $y_i$ are known from previous investigations of thick

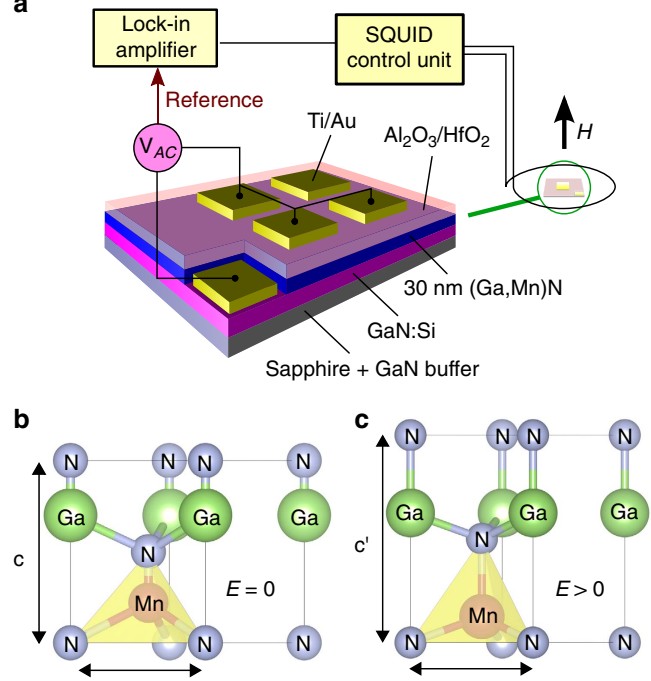

**Figure 2 | Experimental methodology.** (**a**) Schematic illustration of the layer structure and of the circuit used for the measurements. Capacitors containing gate oxide and semi-insulating wurtzite $Ga_{1-x}Mn_xN$ film as insulators are biased by a low-frequency a.c. electric field applied between the gate and the back contact. The structure is placed in the loop of a superconducting quantum interference device (SQUID) magnetometer; changes of the magnetic flux in phase with the electric field are detected by a lock-in amplifier connected to the output of SQUID electronics. (**b,c**) In the case of the $c$ axis parallel to the growth direction and Ga-face polarity, the application of a positive voltage to the gate ($E > 0$) stretches the $Ga_{1-x}Mn_xN$ films along the $c$ axis affecting in this way the single-ion uniaxial magnetic anisotropy.

(relaxed) $Ga_{1-x}Mn_xN$ films with $x = 0.4\%$ (ref. 7), for which the fitting of magnetic anisotropy led to $B_2^0 = 4.2$ meV and $B_4^0 = -0.56$ meV, whereas HRXRD measurements to $a = 3.191$, $c = 5.1863$ Å and, thus, $\xi = -0.0077$.

The lattice parameter $a$ of our thin films is pinned by the GaN buffer. According to X-ray rocking curves and reciprocal space mapping (RSM) $a = (3.1842 \pm 0.0003)$ Å in our samples, and this value is employed further on for the data analysis. In contrast, the minute thickness of our layers precludes a sufficiently precise experimental determination of the $c$ parameter by HRXRD, so that we evaluate it from the magnitude of low-temperature magnetic anisotropy in sample A. This procedure leads to $\xi = -0.0058$.

The corresponding changes of the Jahn-Teller describing the distortion of tetragonal symmetry are unknown but a better fit is obtained by increasing their absolute values by 15% with respect to their magnitudes for $x = 0.4\%$ (ref. 7), that is, to $\tilde{B}_2^0 = -5.9$ meV and $\tilde{B}_4^0 = -1.2$ meV. The spin-orbit parameters are also adjusted in the same range, that is, $\lambda_{TT} = 5.5$ meV, $\lambda_{TE} = 11.5$ meV.

According to data in Fig. 1a,b magnetic anisotropy is weaker in Sample B. As established previously[8], the doping with shallow Si donors reduces the oxidation state of a fraction $z$ of the Mn ions from the $3+$ to the $2+$ charge state, as the $Mn^{2+/3+}$ state resides in the mid gap region. As recalled in the theoretical part of Methods, the magnetization for $Mn^{2+}$ ions in GaN is described

by the isotropic Brillouin function for $S = 5/2$. The magnitudes of $x$, $z$, and $T'$ are determined by fitting the combined theoretical magnetization for $Mn^{3+}$ ions and $Mn^{2+}$ ions with weights $1 - z$ and $z$, respectively, to the whole-set of experimental data.

The most appropriate account of the experimental finding is obtained by taking $x = 2.4\%$ and $z = 0$ for sample A, whereas $x = 2.5\%$ and $z = 0.1$ for sample B. The magnitudes of the effective temperatures $T'$ obtained from the fitting are displayed in Fig. 1. As indicated in Fig. 1 by solid lines, the employed approach with the quoted values of the input parameters provides a correct reproduction of the experimental data.

**Structure processing and experimental set-up.** Having parameterized $M(T, H)$, we turn to the measurements of the magnetic moment as a function of the electric field. In order to minimize the risk of structure damage by an electrical short via, for example, threading dislocations, two precautions are undertaken, as sketched in Fig. 2a. First, after removing by reactive ion etching $(1 \times 1)\,mm^2$ square down to the GaN:Si bottom contact layer, atomic layer deposition is employed to fabricate a 100 nm gate oxide ($Al_2O_3$/$HfO_2$), which after appropriate masking is covered by Ti/Au to serve as a top capacitor plate. Second, a lift-off is used to define over a dozen of independent capacitors with an area smaller than $(1 \times 1)\,mm^2$. The capacitance as well as the dielectric strength and losses are measured for each of these capacitors, and only those showing no leaks are connected in parallel and studied at low temperatures. The magnitude of the resulting capacitance is equal, within the experimental accuracy, to the value expected for oxide and $Ga_{1-x}Mn_xN$ capacitors in series. This allows us to evaluate the magnitude of the electric field $E$ applied vertically to the $Ga_{1-x}Mn_xN$ layer at a given value of the gate voltage $V_G$. We find that this arrangement allows us to apply to $Ga_{1-x}Mn_xN$ an electric field $E$ up to at least $5\,MV\,cm^{-1}$, proving that doping of GaN with Mn results in a device-quality semi-insulating material. For these evaluations we employ the value of the oxide dielectric constant $\epsilon_{ox} = 13$, as determined by capacitance measurements on reference oxide capacitors. We take the dielectric constant of $Ga_{1-x}Mn_xN$ layer to be equal to the one of GaN, $\epsilon_{GaN} = 10$ (ref. 13).

In order to detect directly the effects of an electric field on magnetism we place the structured sample in the loop of a SQUID magnetometer equipped with a home-built head allowing for the electrical connections to be feedthrough down to the

sample chamber. By redirecting the output of the SQUID electronics to the input of a digital lock-in nanovoltmeter, we look for the magnetic signal $M_E$ in phase with the a.c. electric field $E$ of low frequency $f$, as shown in Fig. 2a. Alternatively, the film is biased by a d.c. electric field up to $E = 2.8\,MV\,cm^{-1}$ and the phase sensitive detection of the slope of the $M_E(E)$ is measured with an a.c. component $E_{mod} = 0.26\,MV\,cm^{-1}$.

A noise-limited resolution of $3 \times 10^{-11}$ emu at $f = 1\,Hz$ and for 2 h of lock-in output averaging has been achieved in the absence of magnetic field. Both the signal and noise increase with the magnetic field, but acquisition times up to 2 h ensure an adequate signal to noise ratio under our experimental conditions. No dependence of $M_E$ on $f$ is found between 0.1 and 200 Hz, however $f$ is kept below 3 Hz. The whole set-up is calibrated against a copper coil of known dimensions feeded with an a.c. current. Experiments with the coil demonstrate also the linearity of the output with the magnetic moment over the relevant range of magnetic moments independently of an external magnetic field $H$ up to at least 20 kOe.

**Magnetization as a function of the electric field.** Having calibrated the system, we turn to the experimental determination of the effects of an applied electric field on magnetization. The results obtained at 2 K are summarized in Fig. 3. As indicated, no magnetic signal generated by the electric field is detected in the absence of magnetic field. This is expected as the time reversal symmetry is maintained in the paramagnetic phase and $\alpha = 0$ in the lowest order magnetoelectric term $M_E = \alpha E$ (ref. 14). However, for $H \neq 0$ a sizable electric-field-induced magnetic signal $M_E$ is seen for samples A (all panels in Fig. 3) and sample B (panel **a**). Importantly, this signal is substantially smaller for the easy axis configuration $H \perp c$, as exemplified in Fig. 3b for sample A. Finally, we note that the magnitude of $M_E$ is below the detection limit in the case of sample R (Fig. 3a), which contains more than twenty times less Mn ions than samples A and B.

The magnitude of $M_E$ decays rapidly with increasing temperature, as exemplified in Fig. 4 for sample B. This is an expected behaviour because, according to the comparison presented in Fig. 3c, the PEME signal is directly related to the easy and hard axis magnetization difference of the material, $M_\perp - M_\parallel$, which, on the other hand, has already been shown to vanish at $T \simeq 20\,K$ (Fig. 10 in ref. 7). Additionally, the samples' magnetization response to the electric field gets sizably enhanced at very low temperatures (that is on approaching $T_C \lesssim 1\,K$ in our

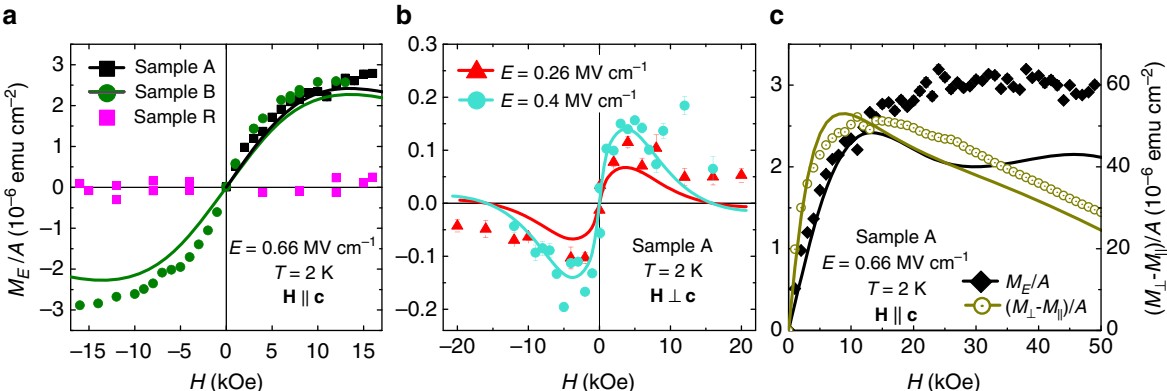

**Figure 3 | Magnetization changes generated by an electric field as a function of magnetic field.** Symbols: experimental data, solid lines: theory. (**a**) Hard axis $H \parallel c$, areal density of the magnetic moment amplitude $M_E/A$ induced by electric field of the amplitude $E = 0.66\,MV\,cm^{-1}$ for all studied samples. (**b**) Easy axis $H \perp c$, response to $E = 0.26$ and $0.4\,MV\,cm^{-1}$ (sample A). (**c**) Response to $E = 0.66\,MV\,cm^{-1}$ for the hard axis $H \parallel c$ and difference in magnetization for the hard and easy axis $(M_\perp - M_\parallel)/A$ in a wide magnetic field range (sample A).

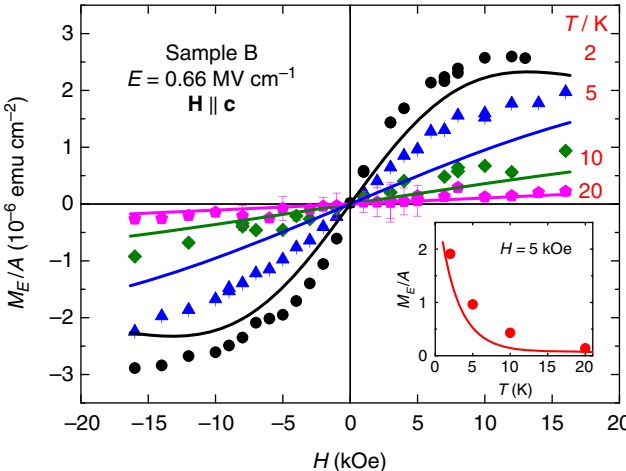

**Figure 4 | Temperature and magnetic-field dependence of magnetization generated by the electric field.** Areal density of the magnetic moment amplitude $M_E/A$ induced by electric field of the amplitude $E = 0.66\,\mathrm{MV\,cm^{-1}}$ (sample B). Inset: temperature dependence in 5 kOe (sample B). Points and solid lines: experimental and theoretical results, respectively.

samples[10]). Indeed, $T'/T \rightarrow 0$ for $T \rightarrow T_C$, as it can be inferred from Fig. 1. The whole temperature trend measured at 5 kOe is followed in the inset to Fig. 4.

According to symmetry considerations, the lowest order non-zero magnetoelectric term is $M_E = \beta H E$. We test this dependency for both experimental configurations. The results obtained at 2 K are displayed in Fig. 5 and indicate that indeed a linear dependence of $M_E$ on $E$ is experimentally observed, such a behaviour is followed both experimentally and theoretically at sufficiently low electric fields. According to Fig. 5a, the linear regime is particularly wide if the experimental values of $M_E$ are determined by integrating the a.c. magnetic response obtained in the presence of a d.c. bias of magnitude $E$ and a weak a.c. modulation of the amplitude $E_{\mathrm{mod}}$. However, $M_E$ obtained directly as a response to the a.c. electric field modulation of amplitude $E$ deviates from the linear dependence for $E > 0.8\,\mathrm{MV\,cm^{-1}}$. The origin of this discrepancy is unclear.

## Discussion

A number of physical mechanisms has been invoked in order to explain magnetoelectric effects in various magnetic systems[15,16]. The proposed models include a variation of the carrier density at the interface, due to gating. Such a change alters either the strength of the exchange coupling between magnetic ions[17] or the occupancy of carrier subbands with a given orientation of orbital momentum[18–20]. Another mechanism, particularly relevant for strong electric fields adjacent to a ferromagnetic metal/oxide interface is the electromigration of atoms, which affects the surface magnetic anisotropy[21]. In magnetic compounds showing ferroelectricity, that is, in multiferroic systems, the electric field leads to an atom rearrangement that modifies the magnetization *via* variations in magnetic anisotropy or spin-spin coupling[2,15,16,22].

We propose here, and support our conjecture with direct quantitative computations that the changes in magnetization generated by the electric field in $\mathrm{Ga_{1-x}Mn_xN}$ films are caused by the inverse piezoelectric effect that alters the lattice parameter $c$ and modifies the trigonal distortion—as shown schematically in Fig. 2b,c—and, thus, modifies the uniaxial magnetic anisotropy.

According to this model, and in agreement with experimental data presented in Fig. 3, the magnitude of magnetization changes induced by the electric field depends on the orientation of the

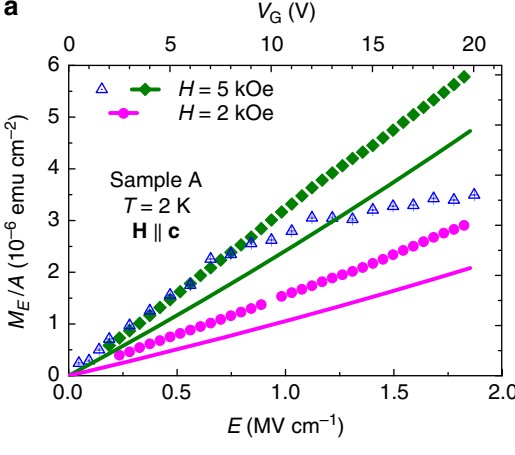

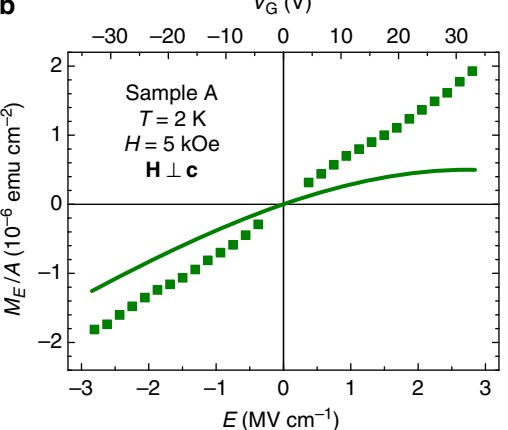

**Figure 5 | Magnetization as a function of the electric field.** (**a**) Areal density of the magnetic moment amplitude induced by electric field $M_E/A$ in the magnetic field along the hard axis is determined in two ways: (i) by applying a low-frequency sinusoidal electric field (open symbols) or (ii) by modulating the d.c. electric field by an a.c. electric field of amplitude $E_{\mathrm{mod}} = 0.26\,\mathrm{MV\,cm^{-1}}$ and displaying the integrated signal (closed symbols). This configuration is investigated at 2 and 5 kOe (sample A). Solid lines: theory. (**b**) Results for the easy axis configuration investigated at 5 kOe by the second method. The upper scales show the gate voltage $V_G$ corresponding to the electric field $E$.

magnetic field in respect to the easy axis. A weak influence of the electric field is expected for the magnetic field along the easy axis, as in such a configuration $M(H)$ follows approximately the Brillouin function for $S = 2$ (see Fig. 1), meaning that it depends only weakly on the trigonal distortion $\xi$ and, thus, on $c(E)$. By contrast, the magnetic response for the hard configuration $\mathbf{H} \parallel \mathbf{c}$ varies strongly with $\xi$, as it is much reduced for $\xi \neq 0$, and tends to the Brillouin function if $\xi \rightarrow 0$, that is, when the uniaxial magnetic anisotropy vanishes.

To quantify all our $E$–dependent results, we adopt the value of the piezoelectric coefficient $d_{33} = 2.8\,\mathrm{pm\,V^{-1}}$, as obtained for a clamped epitaxial GaN film[23] since there is no lateral deformation in thin films deposited on thick buffer layers. Having beforehand parameterized $M(T, H)$, as reported in Fig. 1, and knowing $c(E)$ and, thus, $\xi(E)$ we are in the position to determine $M_E(T, H)$ with no additional adjustable parameters. Theoretical results obtained in this way are represented by solid lines in Figs 3–5.

As seen, the model proposed here gives a good account of all experimental results as a function of the three experimentally controllable parameters: $\mathbf{H}$, $T$ and $E$. The level of achieved agreement indicates that the PEME dominates in the studied

$Ga_{1-x}Mn_xN$ films. At the same time, it is clear that in some cases there is only qualitative agreement between experimental results and theoretical expectations (see, for example, high–field data in Fig. 3c, at 5 K in Fig. 4, and in high electric fields in Fig. 5). On the experimental side, as already noted, surface roughness detected by AFM may lead to lateral non-uniformities of the applied electric field, which can affect the behaviour of the magnetoelectric effect. It is also possible that the observed discrepancies indicate that the piezoelectric stretching not only changes the magnitude of trigonal distortion but also alters, for example, the Jahn-Teller effect or spin-spin coupling. Finally, the discrepancies may herald an onset of a contribution from a not yet identified phenomenon beyond PEME.

To put our findings in a more general perspective, it is worth recalling that dilute ferromagnetic semiconductors, such as $Ga_{1-x}Mn_xAs$, have been playing a major role in seeding new concepts of spintronic devices and in developing our quantitative understanding of the interplay between magnetic and semiconducting properties[12,24,25]. Among these concepts particularly far reaching is the demonstration of magnetization control by an electric field[17,19]. In the present paper, this methodology has been extended to the wurtzite dilute magnetic insulator $Ga_{1-x}Mn_xN$, in which the Fermi energy is pinned by Mn ions in the mid-gap region and the $Mn^{3+}$ ions show strong single-ion anisotropy. By applying a new detection scheme, we have demonstrated magnetization control by the electric field *via* the inverse piezoelectric effect and developed a theory describing the PEME quantitatively. In this way, our work bridges two fields of research developed so far independently: piezoelectricity of wurtzite III–V and II–VI semiconductors[1] and electrical control of magnetization in hybrid and composite magnetic structures containing piezoelectric components[2–5].

According to the results presented here, the PEME shows a strong increase for $T \to T_C$. Since $T_C$ grows rapidly with $x$ in $Ga_{1-x}Mn_xN$ ($T_C \propto x^{2.2}$, refs 10,26), a further progress in the incorporation of Mn into GaN will shift the magnetization control by an electric field towards higher temperatures. It is worth mentioning in this context that in addition to the influence of the inverse piezoelectric effect upon single-ion magnetic anisotropy dominating in the case considered here, an impact of the electric field upon spin–spin interactions may dominate at higher Mn concentrations. While we have detected magnetization changes by direct magnetization measurements, spin-dependent tunnelling[27], spin-Hall magnetoresistance[28] and magnetooptical phenomena[29] may also serve to probe magnetization in magnetic insulators.

Another worthwhile outcome of the present work is the notion that the magnitude of the PEME, or more pictorially the depth of the magnetic anisotropy modulation, which depends on $E$ and on the $d_{33}$ coefficient of the given material, will be the stronger the larger is the $\Delta\xi(E) = \xi(E) - \xi(0)$ change with respect to $\xi(0)$. An inversion of the magnetic anisotropy of the material (from in-plane to perpendicular, or *vice versa*) is expected when

$\xi$ changes sign, $\xi(E)\xi(0) < 0$. As for many materials small magnitudes of dielectric strength or $d_{33}$ may preclude obtaining a sufficiently large $\Delta\xi(E)$, advancements in material science may allow to engineer such a strain in layers with $\xi(0) \cong 0$ and so the condition $\xi(E)\xi(0) < 0$ could be met and the easy magnetization axis switching could be realized with accessible electric fields. For the $Ga_{1-x}Mn_xN$ films studied here a use of compressive strain generated by an $Al_xGa_{1-x}N$ buffer would be a viable option.

Finally, we note that according to our results Mn doping is the method to turn GaN into a high-quality semi-insulating material that sustains a strong electric field. Importantly, Mn doping does not deteriorate structural properties, as shown in Methods, up to Mn concentrations $x \approx 3\%$ achievable by MOVPE. This is in contrast to other transition metals, such as Fe (ref. 30) and Cr (ref. 31) that tend to aggregate into transition metal-rich nanocrystals for $x \gtrsim 0.5\%$ (refs 32,33), the process destroying their carrier trapping capabilities[32]. Similarly, heavy doping with C acceptors results in the dislocation formation and the appearance of self-compensating defects[30,34].

## Methods

**Atomic force microscopy.** The GaN and $Ga_{1-x}Mn_xN$ layers co-doped with Si are grown in an AIXTRON 200RF horizontal tube MOVPE reactor. Information on the morphology of the sample surface is obtained from tapping-mode AFM performed with a VEECO Dimension 3100 AFM system. The $(40 \times 40)$ $\mu m^2$ area scans for Samples A, B and R are reported in Fig. 6. The root mean square surface roughness for samples A and B doped with a nominal Mn concentration of 3%, is 7.9 and 6.8 nm, respectively. For sample R with Mn concentration 0.1%, the surface roughness is estimated to be 5.5 nm.

**High-resolution x-ray diffraction.** HRXRD measurements have been carried out in a PANAlytical X'Pert PRO materials research diffractometer equipped with a hybrid monochromator with a 1/4° divergence slit. The diffracted beam is collected by a solid state PixCel detector with a 9.1 mm antiscatter slit. Radial $2\theta - \omega$ scans between 30° to 80° are acquired along the growth direction of GaN [0001], covering the symmetric (0002) and (0004) Bragg diffraction peaks and are shown in Fig. 7a–c. The $2\theta - \omega$ scans confirm the high crystallinity of the epitaxial layers with no precipitation of secondary phases such as $Mn_4N$ or SiN. The lattice parameters are calculated from rocking curve measurements of the symmetric (002) and asymmetric ($-105$) Bragg peaks. Furthermore, the full-width half maxima of the symmetric (0002) and asymmetric ($-1015$) reflections are estimated to be 150–160 arcsec.

RSM—that is, the scanning of a two-dimensional region of a crystal in the reciprocal space—provide information on interplanar spacings and on the effect of structural defects on the crystal arrangement. The intensity in a RSM is the projection of the three-dimensional intensity of the diffraction reflexes on a two-dimensional plane. In the RSMs reported in Fig. 7d–f, for the measured samples A, B and R, the broadening in the $Q_x$ direction is assigned to sample tilt, finite lateral coherence length of mosaic blocks and to compositional fluctuations. The absence of broadening of the peak intensity along the $Q_z$ direction indicates that the considered $Ga_{1-x}Mn_xN$ and GaN:Si layers are strained on the GaN buffer.

**Secondary-ions mass spectrometry.** The SIMS measurements have been performed in a Cameca IMS 6 F using either $^{133}Cs^+$ or $^{16}O_2^+$ ions, and are presented in Fig. 8 for samples A, B and R. The profiles confirm the nominal thickness of the Mn-containing layers (30 nm) and are consistent with the Mn concentration established from the saturation moment at 2 K (2.4 and 2.5% for samples A and B, respectively, as given in the main text).

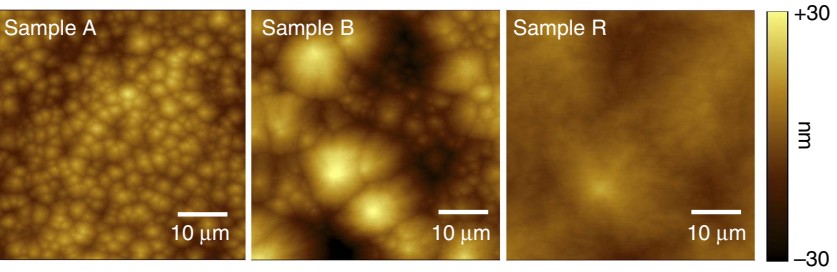

**Figure 6 | Atomic force microscopy surface imaging.** Tapping mode images for all three studied layers.

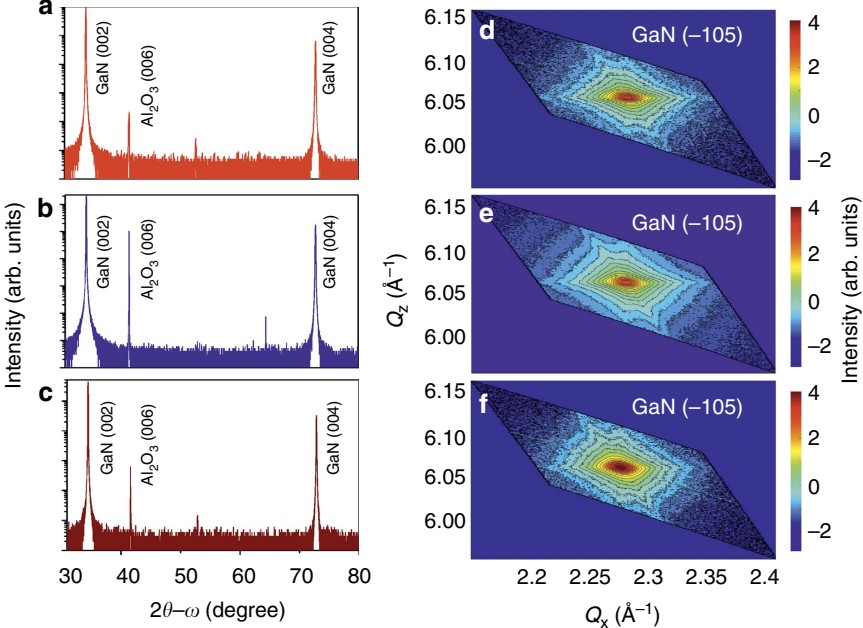

**Figure 7 | High-resolution X-ray diffraction data.** (**a**–**c**) Radial $2\theta - \omega$ scans of symmetric Bragg reflections for Samples A, B and R, respectively. (**d**–**f**) Reciprocal space maps of asymmetric ($-105$) Bragg peak for Samples A, B and R.

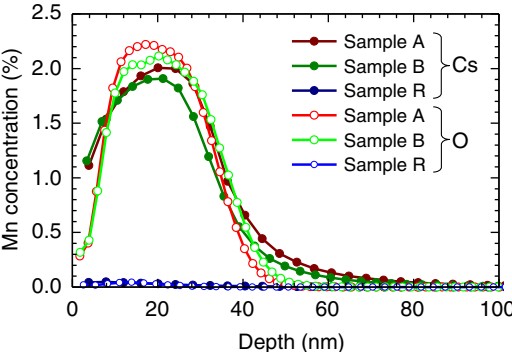

**Figure 8 | Secondary-ions mass spectrometry Mn-depth profiles.** The measurements are performed separately using either $^{133}Cs^+$ (full symbols) or $^{16}O_2^+$ ions (open symbols).

**High-resolution transmission electron microscopy.** HRTEM measurements have been carried out employing a FEI Titan Cubed 80–300 scanning transmission electron microscope (STEM). High-angle annular dark-field images (top row of Fig. 9) show a contrast between GaN and the top $Ga_{1-x}Mn_xN$, whose thickness is evaluated to be 32 nm for both samples, in a good agreement with the nominal value of 30 nm. No hint of Mn aggregation is detected. HRTEM images (bottom row in Fig. 9), similarly to HRXRD, demonstrate the absence of crystalline phase separation in the studied layers.

**SQUID magnetization measurements.** Among the many factors limiting the credibility of magnetometry in nanomagnetism, exhaustively discussed previously[11], the presence of bulky substrates with a magnetic moment typically larger than those of the measured films is the most detrimental one. At first glance, the most appropriate approach calls for a direct subtraction of the magnetic response of the clear substrate at the matching fields and temperatures and then scaled according to its mass. However, such procedure yields a proper result only when the investigated sample and a reference have commensurate shapes as, actually, the real moment of the measured specimen is $m(T, H)/\gamma$, where $\gamma \simeq 1$ is the coupling factor of a uniformly magnetized body with finite pick-up coils of a SQUID magnetometer. For a point object $\gamma = 1$. Table 1 in ref. 11 lists $\gamma$ values for some of the most typical sample shapes, indicating that actually it is the substrate's highly nonsymmetrical shape that introduces the most significant error in the magnetic anisotropy data. Indeed, $(\gamma_\parallel - \gamma_\perp)/\gamma_\parallel \approx 5\%$ for a typical tile shaped samples, introducing an error easily exceeding the magnitude of the investigated anisotropy, particularly in strong magnetic fields.

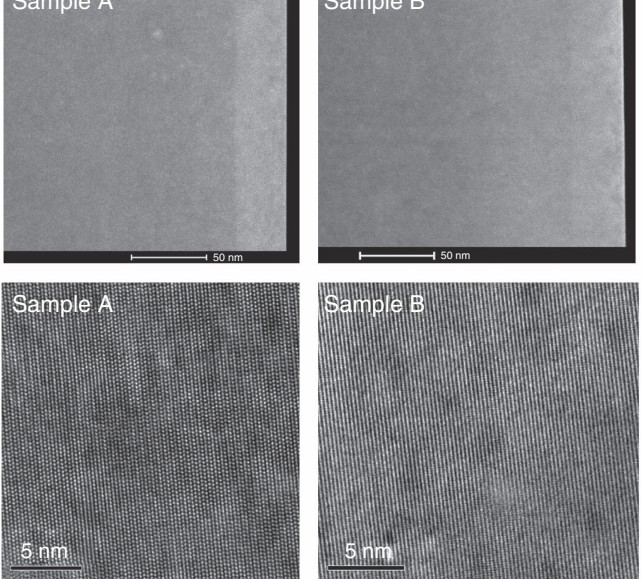

**Figure 9 | High-resolution transmission electron microscopy images of samples A and B.** Top row: high-angle annular dark-field/ scanning transmission electron microscopy (HAADF/STEM), a contrast between GaN and 32-nm thick (Ga,Mn)N films on the top of the structures is detected with no indications of Mn aggregation. Bottom row: high-resolution transmission electron microscopy (HRTEM) images for both layers reveal their single phase nature.

In order to mitigate this challenge, both the original $(5 \times 5 \times 0.3)$ mm$^3$ samples and a piece of matching sapphire substrate have been diced into four $\sim 1.2$ mm wide strips and glued to form a cuboid of a square cross-section, $(1.2 \times 1.2 \times 5)$ mm$^3$, as illustrated in Fig. 10. In this way, both in-plane and perpendicular measurements can be carried out upon a simple rotation of the specimen by 90° along its length, thus preserving $\gamma$, and using the same sample holder, the procedure assuring the required reproducibility of the experimental conditions.

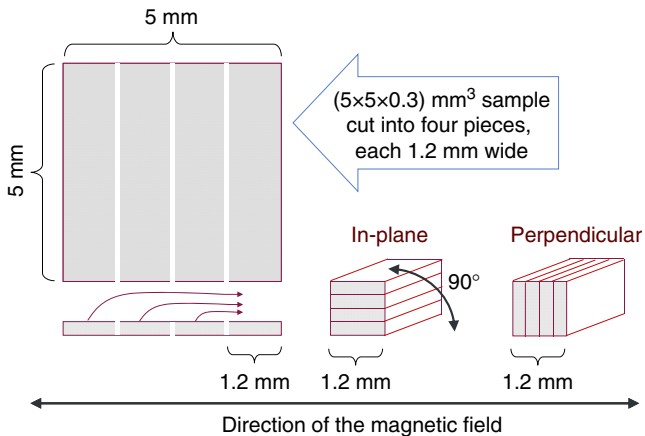

**Figure 10 | Experimental procedure for measuring magnetic anisotropy in thin films.** An illustration of a modification applied to a standard (here: square) sample resulting in a shape invariable with respect to 90° rotations required for setting 'in-plane' and 'perpendicular' experimental configurations during magnetic anisotropy studies.

**Theoretical model and relevant material parameters.** In the case of $Mn^{2+}$ ions for which the ground state is the orbital singlet ($S = 5/2$, $L = 0$) in $Ga_{1-x}Mn_xN$ the magnitude of magnetization is given by the Brillouin function $B_S$,

$$M(T,H) = g\mu_B N_0 x_{Mn2+} S B_S[g\mu_B H/k_B T], \qquad (1)$$

where $g = 2.0$, $N_0 = 44.1\ nm^{-3}$ is the cation concentration, $x_{Mn2+} = 0$ for sample A and $0.1x$ for sample B, and $S = 5/2$.

Because of a non-zero value of the orbital momentum, the description of magnetization generated by $Mn^{3+}$ ions is more complex. According to the group theory the energy structure of a single $Mn^{3+}$ ion in wurtzite GaN can be described by the Hamiltonian[6,7,35],

$$\mathcal{H} = \mathcal{H}_{CF} + \mathcal{H}_{JT} + \mathcal{H}_{TR} + \mathcal{H}_{SO} + \mathcal{H}_Z, \qquad (2)$$

where $\mathcal{H}_{CF} = -2/3 B_4(\hat{O}_4^0 - 20\sqrt{2}\hat{O}_4^3)$ describes the crystal filed of the tetrahedral $T_d$ symmetry, $\mathcal{H}_{TR} = B_2^0\hat{O}_4^0 + B_4^0\hat{O}_4^2$ corresponds to the trigonal distortion along the GaN hexagonal $c$ axis, which lowers the symmetry to $C_{3V}$, $\mathcal{H}_{JT} = \tilde{B}_2^0\hat{\Theta}_4^0 + \tilde{B}_4^0\hat{\Theta}_4^2$ gives the static Jahn-Teller distortion of the tetragonal symmetry, $\mathcal{H}_{SO} = \lambda\hat{L}\hat{S}$ represents the spin–orbit interaction, and $\mathcal{H}_Z = \mu_B(\hat{L} + 2.0\hat{S})\mathbf{H}$ is the Zeeman term. Here $\hat{\Theta}$, $\hat{O}$ represent Stevens equivalent operators for a tetragonal distortion along one of the cubic [100] axes and the trigonal axis [111]∥$c$, respectively. In general, to take into account the fact that the hybridization of the $d$ wave function with the ligand wave function is different for the $^5T_2$ and $^5E$ states, three different spin-orbit parameters: $\lambda_{TT}$, $\lambda_{TE}$, and $\lambda_{EE} = 0$ are introduced[6,35].

The ground state of the $Mn^{3+}$ ion is an orbital and spin quintet $^5D$ with $L = 2$ and $S = 2$. The term $H_{CF}$ splits the $^5D$ ground state into two terms of symmetry $^5E$ and $^5T_2$ (ground term). The $^5E - {}^5T_2$ splitting is $\Delta_{CF} = 120B_4$. The nonspherical $Mn^{3+}$ ion undergoes further Jahn-Teller distortion, that lowers the local symmetry and splits the ground term $^5T_2$ into an orbital singlet $^5B$ and an higher located orbital doublet $^5E$. The trigonal field splits the $^5E$ term into two orbital singlets and slightly decreases the energy of the $^5B$ orbital singlet. The spin-orbital term yields further splitting of the spin orbitals. Finally, an external magnetic field lifts all of the remaining degeneracies.

For the crystal under consideration, there are three Jahn-Teller directions: [100], [010] and [001] (center $A$,$B$,$C$, respectively)[6,35].

The energy level scheme of the $Mn^{3+}$ ion is calculated through a numerical diagonalization of the full $25 \times 25$ Hamiltonian (2) matrix. The resulting magnetization assumes then the form,

$$M(T,H) = \mu_B N_0 x_{Mn3+} Z^{-1} \\ \cdot (Z_A\langle\mathbf{m}\rangle^A + Z_B\langle\mathbf{m}\rangle^B + Z_C\langle\mathbf{m}\rangle^C), \qquad (3)$$

where $x_{Mn3+} = x$ for sample A and $0.9x$ for sample B, $Z_i$ ($i = A$, $B$ or $C$) are the partition function of the $i$-th center, $Z = Z_A + Z_B + Z_C$, and

$$\langle\mathbf{m}\rangle^i = -\frac{\sum_{j=1}^N \left\langle \varphi_j|\hat{L} + 2.0\hat{S}|\varphi_j\right\rangle \exp\left(-E_j^i/k_B T'\right)}{\sum_{j=1}^N \exp\left(-E_j^i/k_B T'\right)}, \qquad (4)$$

where $E_j^i$ and $\varphi_j$ are the $j$-th energy level and the eigenstate of the $i$-th center, respectively, and $T' = T - T_0(T)$ is a fitting parameter taking into account ferromagnetic interactions between $Mn^{3+}$ ions.

Although, the values of the crystal field, Jahn-Teller and spin-orbit parameters for wurtzite $Ga_{1-x}Mn_xN$ can be found in literature they are specific either to strongly compensated bulk crystals of very low Mn content but high O and Mg concentrations[6,35] or to thick (unstrained) epitaxial layers of significantly smaller Mn contents[7] than in our case. Therefore, as already pointed out in the main text, the parameter values are adjusted to reproduce the initial magnetization data (Fig. 1) for epitaxially strained epilayers with a relatively high Mn concentrations studied in our work. As explained in the main text, a particularly sizable modification is expected for the parameters $B_2^0$ and $B_4^0$, as they describe the trigonal deformation, so that they are linear in $\xi = c/a - \sqrt{8/3}$, $B_i^0 = y_i\xi$. Namely, we use: $B_4 = 11.44$ (11.44) meV $\xi = -0.0058$ ($-0.0077$), $y_2 = -550$ ($-550$) meV $y_4 = 73$ (73) meV, $\tilde{B}_2^0 = -5.9$ ($-5.1$) meV, $\tilde{B}_4^0 = -1.2$ ($-1.02$) meV, $\lambda_{TT} = 5.5$ (5.0) meV, $\lambda_{TE} = 11.5$ (10.0) meV, and $\lambda_{EE} = 0$ (0), where the numbers in parentheses correspond to the very dilute $Ga_{1-x}Mn_xN$ unstrained epilayers[7].

In calculations of the PEME, the value $\xi(E)$ is obtained from $[c(E) - c(0)]/c(0) = Ed_{33}$, where $d_{33} = 2.8\ pmV^{-1}$ (ref. 23) and $c(0)$ is determined from $\xi(0) = -0.0058$ and the clamped value of $a = 3.1842$ Å, known from the HRXRD measurements.

**Data availability.** The data that support the findings of this study are available from the corresponding author upon request.

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

## Acknowledgements

We thank Elżbieta Łusakowska for some of AFM measurements. The work is supported by the Narodowe Centrum Nauki (Poland) through projects MAESTRO (2011/02/A/ST3/00125), OPUS (2013/09/B/ST3/04175) and FUGA (2014/12/S/ST3/00549), by the Austrian Science Foundation—FWF (P22477, P24471 and P26830), by the NATO Science for Peace Programme (Project no. 984735), and by the European Commission through the 7th Framework Programmes: CAPACITIES project REGPOT-CT-2013-316014 (EAgLE), and Horizon2020 Project NMP645776 ALMA.

## Author contributions

The work was planned and proceeded by discussions among A.B., M.S., and T.D. Samples growth by MOVPE and their structural characterization by AFM and HRXRD were carried out by R.A. and A.B. Atomic layer deposition and RIE were performed by K.K., R.K., and A.P., whereas R.J. and T.L. carried out SIMS and TEM studies, respectively. K.G. and M.S. performed SQUID magnetization measurements, whereas D.S., M.F., G.P.M. and M.S. processed the samples and made SQUID studies in the electric field with an input of M.Z. Analysis and interpretation of the data were accomplished by D.S. and T.D. The manuscript was written by D.S., M.F., G.P.M., A.B., M.S., and T.D. with inputs from all the authors.

## Additional information

**Competing financial interests:** The authors declare no competing financial interests.

**Publisher's note**: 

