## [Peer Review File · Nature Communications]

Reviewers' Comments:

Reviewer #1 (Remarks to the Author):

The paper reports interesting findings on magnetic properties in GaMnN in relation to electric fields. The paper is original and the data and methodology clear. The authors assume that Mn is incorporated on specific sites, but neglect the ongoing discussion about the possibilities of Mn clusters in GaN as well as the possibility that Mn ions may sit on random locations as was found for Cu incorporation in GaN which also yielded ferromagnetic behaviour.

I recommend that the authors address issues related to Mn clustering which might affect magnetic properties as well as the effects of external electrical fields as a possible revision.

Reviewer #2 (Remarks to the Author):

The authors show that Mn-doped GaN exhibits an interesting magneto-electric effect. The first paragraph claims to make three main conclusions, which gives a sense of inflation. The main claim is the second one, i.e., GaMnN exhibits a magneto-electric effect. The first claim clarifies that this magneto-electric effect is piezoelectric in nature, and the third claim of "device quality semi-insulating material" is an interesting observation on the side, but should not really be put forward as a main conclusion.

The discovery that GaMnN exhibits a piezo-electro-magnetization effect seems to be new and original. The authors clarify that such effects are known in hybrid or nanocomposite systems, but not in a single compound. I assume the AFM, HRXRD, SIMS, and HAADF/STEM data is given to convince the reader that we are dealing with a genuine (single crystal?) "single compound", and not a hybrid or nanocomposite system, which could be the case if Mn exhibits strong phase separation tendencies (spinodal decomposition). A lot of the interpretation is left to the reader. The authors should be more clear on why the various experimental data are given, and present concise and clear conclusions that relate each of these data to the main claims of the paper. Omitting any of the presented data would leave open important questions of the sample quality, and clearly is not an alternative.

The theoretical curves in Figs 3-5 seem to fit quite well, but reproducing them "without adjustable parameters" appears non-trivial. Clearly the model contains quite a lot of parameters, starting from "introducing an effective temperature as a fitting parameter", assuming that 10% of the Mn impurities (only in sample B?) are in the 2+ charge state, as well as adjusting the trigonal deformation parameter, Jahn-Teller parameters, spin orbit parameters, and so on. Once all of these adjustments are done, indeed, the authors seem to be in the position to determine $M_E(T,H)$, but verifying all of this would require more time than is given to the referee to review the paper. Surely all the parametrized formulas for magnetization curves, anisotropies, etc are given somewhere in the references cited. The relevant parametrized formulas with brief motivations should also be given in the Methods section. Otherwise the claim of having developed a theory (last line of abstract) is too implicit.

The fact that GaMnN exhibits a magneto-electric effect, even if only at 2K, is interesting. The experimental work and theory development seem reliable, but too much interpretation is left to the reader. This makes it difficult to judge their validity, impact and importance. The paper is quite hard to read.

For specialists studying GaMnN this paper is interesting, and it may influence their thinking.

Reviewer #3 (Remarks to the Author):

In this manuscript, authors reported the observation of piezoelectro-magnetization effect (PEME) in a famous diluted magnetic semiconductor, (Ga,Mn)N. As stated by authors, this is the first time that two research fields, piezoelectricity and magnetoelectric effect, have been bridged up in a single wurtzite compound instead of other hybrid and nanocomposite systems. This work is of general interests to the community of both piezoelectricity and DMS, and enlarges the potential application of spintronics. This work is important, and I believe that "it will influence the thinking in the field". I support it to be published in NC with the considerations of the following suggestions:

1, The procedure and parameters used for theoretical fit should be explained more in detail.

2, I would suggest authors to give some explanations for Fig. 1 where hysteresis loop are observed at 2 K for both (H//c and H perpendicular to c) while T_C is only 1 K for this sample. This will help non-experts to follow the idea.

3, Some parameters such as $B^0_2 = 3.2$ mV (on page 2), and spin-orbit parameters $\lambda_{\{TT\}}$ etc. should be introduced and explained in detail.

4, It seems to me that the theoretical fit to experimental data in Fig. 3c for H//c is not good, especially for $H > 15$ KOe. More explanations are welcomed. Actually, the fit to 5 K data in Fig. 4 is not good as well.

5, Typo: On page 1, the first paragraph, " First, the magnetoelectric effect generated by piezoelectricity can exist is a single compound". Here "is" should be "in".

We would like to thank the Reviewers for their time and expertise devoted to our manuscript as well as for positive opinions about our work. According to Reviewer #1: *“The paper reports interesting findings on magnetic properties in GaMnN in relation to electric fields. The paper is original and the data and methodology clear.”* Similarly Reviewer #2 stated: *“The discovery that GaMnN exhibits a piezo-electro-magnetization effect seems to be new and original.”* These opinions are shared by Reviewer #3 who wrote: *“This work is of general interests to the community of both piezoelectricity and DMS, and enlarges the potential application of spintronics. This work is important, and I believe that “it will influence the thinking in the field”. I support it to be published in NC”*

At the same the Reviewers put forward well-taken and constructive criticisms. We provide here a point-by-point reply to Reviewers’ comments and list the corresponding modifications introduced to the revised manuscript. Furthermore, we have prepared a manuscript in which revised sentences are marked in blue, whereas new sentences in red.

Reviewer #1:

#R1) The authors assume that Mn is incorporated on specific sites, but neglect the ongoing discussion about the possibilities of Mn clusters in GaN as well as the possibility that Mn ions may sit on random locations as was found for Cu incorporation in GaN which also yielded ferromagnetic behaviour. I recommend that the authors address issues related to Mn clustering which might affect magnetic properties as well as the effects of external electrical fields as a possible revision.

Our reply:

We agree entirely with the Reviewer#1 that issues related to the possibility of the presence/formation of Mn-rich clusters is absolutely crucial. Our previous works (e.g., Refs 6, 8, 12) show that we have been always aware of the issue and over the years developed a protocol of in-depth characterization aiming exactly at establishing the nature of the incorporation of transition metal ions in a semiconducting matrix. In order to respond to the grounded concern of the Reviewer, we have added to the original text the statements as follows :

1) p. 2, subsection “Samples”, the new paragraph reads:

“Details on the instrumentation and on the results of the comprehensive structural and chemical characterization of the samples are collected in Methods. Both HRXRD and HRTEM images point to a high crystalline quality of the wz structures as well as reveal neither secondary phases nor relaxation defects in strained Ga_{1-x}Mn_xN layers. According to the high angle annular dark-field TEM images, Ga_{1-x}Mn_xN layer thickness is $t = 32$ nm with no indication of chemical phase separation (Mn aggregation) that might be driven by spinodal decomposition. This magnitude of t is consistent with SIMS data that corroborate also the values of Mn concentrations deduced from magnetization measurements on samples A, B, and R discussed in the next subsection. The magnetization data confirm also that the Mn distribution is uncorrelated and that an overwhelming majority of Mn ions assumes the 3+ charge state specific to the Ga-substitutional case.”

2) p. 3, subsection “Magnetization as a function of the magnetic field”, new paragraph reads:

“The circles in Fig. 1 represent the experimental data $M(T,H)$ per unit sample area A for the A and B Ga_{1-x}Mn_xN films at two orientations of the magnetic field H with respect to the c -axis. No hysteretic behaviour is found down to the experimental noise level $[\sigma_M/A(H \approx 0)] \leq 1 \times 10^{-6}$ emu/cm², where $A \approx 0.25$ cm²] at any investigated temperature (2, 5, 10 K – presented in Fig. 1 -- and at 50 and 300 K, not shown). This points to the absence of a ferromagnetic phase that might originate from Mn-rich

aggregates. This conclusion is further supported by results of HRXRD and HRTEM studies on these samples presented in Methods, and which -- within the ultimate limits specific to these techniques -- rule out the presence of any Mn-rich phases either in the form of nanometric precipitates or of regions of spinodal decomposition¹²."

- 3) p. 7, Methods, subsection "High resolution x-ray diffraction"; the relevant statement describing data presented in Fig. 7 is modified to the form:

"The 2θ - ω scans confirm the high crystallinity of the epitaxial layers with no precipitation of secondary phases such as Mn_4N or SiN ."

- 4) p. 9, Methods, subsection "High-resolution transmission electron microscopy"; the relevant statement is modified to the form:

"High-angle annular dark-field (HAADF) images (top row of Fig. 9) show a contrast between GaN and the top $Ga_{1-x}Mn_xN$, whose thickness is evaluated to be 32 nm for both samples, in a good agreement with the nominal value of 30 nm. No hint of Mn aggregation is detected. HRTEM images (bottom row in Fig. 9), similarly to HRXRD, demonstrate the absence of crystalline phase separation in the studied layers."

Reviewer #2:

#R2a) The authors show that Mn-doped GaN exhibits an interesting magneto-electric effect. The first paragraph claims to make three main conclusions, which gives a sense of inflation. The main claim is the second one, i.e., GaMnN exhibits a magneto-electric effect. The first claim clarifies that this magneto-electric effect is piezoelectric in nature, and the third claim of "device quality semi-insulating material" is an interesting observation on the side, but should not really be put forward as a main conclusion.

Our reply:

Following the Referee's recommendation the whole sentence related to the third claim of "device quality semi-insulating material" was removed from the first paragraph, which now reads:

"...the findings presented in this Communication lead to the following conclusions. (i) the magnetoelectric effect generated by piezoelectricity can exist in a homogeneous crystalline compound, not only in hybrid or nanocomposite systems; (ii) the multifunctional capabilities of $Ga_{1-x}Mn_xN$ extend into the core of spintronic functionalities, like the manipulation of magnetization by electric field."

#R2b) The discovery that GaMnN exhibits a piezo-electro-magnetization effect seems to be new and original. The authors clarify that such effects are known in hybrid or nanocomposite systems, but not in a single compound. I assume the AFM, HRXRD, SIMS, and HAADF/STEM data is given to convince the reader that we are dealing with a genuine (single crystal?) "single compound", and not a hybrid or nanocomposite system, which could be the case if Mn exhibits strong phase separation tendencies (spinodal decomposition).

Our reply:

As this objection is related to the incorporation of Mn into our layers, and it is essentially identical to the comment of Reviewer #1, we kindly ask Reviewer #2 to refer to our reply #R1 given above.

R#2c) A lot of the interpretation is left to the reader. The authors should be more clear on why the various experimental data are given, and present concise and clear conclusions that relate each of these data to the main claims of the paper. Omitting any of the presented data would leave open important questions of the sample quality, and clearly is not an alternative.

And later: **This makes it difficult to judge their validity, impact and importance. The paper is quite hard to read.**

Our reply:

To address this criticism, in addition to rewording quite a few of sentences, we have decided to add to the introductory section (pp. 1 and 2) four new paragraphs that describe the paper logic, making -- we hope -- the paper more readable. These new paragraphs read:

“Our paper consists of four major parts. First, we discuss the layout of our samples and demonstrate, by employing a range of nanocharacterization tools and SQUID magnetometry, that the prepared structures show the assumed architecture and that the $Ga_{1-x}Mn_xN$ layers are of good crystalline quality and contain randomly distributed Ga-substitutional Mn ions.

The second part contains results of magnetization measurements as a function of magnetic field at different temperatures and field orientations in respect to the c-axis of the wz structure. The data show that the studied samples are in the paramagnetic phase in the relevant temperature range, $T \leq 2$ K. Theoretical interpretation of the magnetization results updates the values of parameters characterizing the magnitudes of the crystal field, spin-orbit coupling, and Jahn-Teller effect of Mn^{3+} ions to the relevant case of strained $Ga_{1-x}Mn_xN$ films with $x \approx 2.5\%$ deposited epitaxially on a thick GaN buffer layer.

In the third part we describe processing steps leading to a relatively large structures amp to sustain high electric fields. An experimental set-up allowing for a direct detection of magnetization changes generated by the electric field is also described.

The fourth key part of the paper presents experimentally detected changes of magnetization generated by the applied electric field as a function of magnetic field, electric field, and temperature. Adopting the literature value of the piezoelectric coefficient determined for GaN and the set of the parameters employed to describe magnetization we reproduce by our theory the magnitude of the piezoelectro-magnetization effect with no additional adjustable parameters.”

Furthermore, in order to explain why various experimental data were given (as rightly recommended by the Reviewer) we have supplemented the subsection “Samples” by a new paragraph (p. 2). This additional paragraph follows:

“Details on the instrumentation and on the results of the comprehensive structural and chemical characterization of the samples are collected in Methods. Both HRXRD and HRTEM images point to a high crystalline quality of the wz structures as well as reveal neither secondary phases nor relaxation defects in strained $Ga_{1-x}Mn_xN$ layers. According to the high angle annular dark-field TEM images, $Ga_{1-x}Mn_xN$ layer thickness is $t = 32$ nm with no indication of chemical phase separation (Mn aggregation) that might be driven by spinodal decomposition. This magnitude of t is consistent with SIMS data that corroborate also the values of Mn concentrations deduced from magnetization measurements on samples A, B, and R discussed in the next subsection. The magnetization data confirm also that the Mn distribution is uncorrelated and that an overwhelming majority of Mn ions assumes the 3+ charge state specific to the Ga-substitutional case.”

Furthermore, in order to address the critics: **“too much interpretation is left to the reader”** we have revised the manuscript considerably, especially the parts devoted to modelling of our experimental results. In particular, we have added in Methods a new subsection “Theoretical model and relevant

material parameters” (pp 9, 10), in which the relevant Hamiltonians are shown and parameters involved discussed. Furthermore, we have rewritten the subsection “Magnetization as a function of the magnetic field”, in particular all paragraphs relevant in the context of numerical fitting of the experimental data. These reedited paragraphs begins by the statement: “*The starting point of the theory aiming in description of $M(T,H)$...*” (p. 3).

Further modifications of the manuscript, also -- we hope -- explaining better the interpretation of our results to the readers have been introduced in reaction to the next comment of the Reviewer.

R#2d) The theoretical curves in Figs 3-5 seem to fit quite well, but reproducing them "without adjustable parameters" appears non-trivial. Clearly the model contains quite a lot of parameters, starting from "introducing an effective temperature as a fitting parameter", assuming that 10% of the Mn impurities (only in sample B?) are in the 2+ charge state, as well as adjusting the trigonal deformation parameter, Jahn-Teller parameters, spin orbit parameters, and so on. Once all of these adjustments are done, indeed, the authors seem to be in the position to determine $M_{_E}(T,H)$, but verifying all of this would require more time than is given to the referee to review the paper. Surely all the parametrized formulas for magnetization curves, anisotropies, etc are given somewhere in the references cited. The relevant parametrized formulas with brief motivations should also be given in the Methods section. Otherwise the claim of having developed a theory (last line of abstract) is too implicit.

Our reply:

Agreeing with the Referee comment “**The theoretical curves in Figs 3-5 seem to fit quite well, but reproducing them ‘without adjustable parameters’ appears non-trivial. Clearly the model contains quite a lot of parameters**” we have withdrawn “*no adjustable parameters*” in the abstract and in the main text we have replaced “*no adjustable parameters*” by “*no additional adjustable parameters*” when referring to results presented in Figs 3-5 (which show electric field effects).

Furthermore, following Reviewer’s recommendation, as mentioned in our reply to the previous comment we have added in the Methods section a new subsection “Theoretical model and relevant material parameters” (pp 9, 10) and in the main text rewritten the subsection “Magnetization as a function of the magnetic field” (p. 3). In this part a new statement addressing the question of Mn 2+ charge state in Sample B is introduced after discussing the interpretation of magnetization $M(T,H)$ for Sample A. This statement reads:

“According to data in Figs 1a and 1b magnetic anisotropy is weaker in Sample B. As established previously⁶, the doping with shallow Si donors reduces the oxidation state of a fraction z of the Mn ions from the 3+ to the 2+ charge state, as the $Mn^{2+/3+}$ state resides in the mid-gap region. As recalled in the theoretical part of Methods, the magnetization Mn^{2+} ions in GaN is described by the isotropic Brillouin function for $S = 5/2$. The magnitudes of x , z , and T are determined by fitting the combined theoretical magnetization for Mn^{3+} and ions and Mn^{2+} ions with weights $1 - z$ and z , respectively to the whole set of experimental data.”

Reviewer #3:

This work is important, and I believe that "it will influence the thinking in the field". I support it to be published in NC with the considerations of the following suggestions:

R#3-1) The procedure and parameters used for theoretical fit should be explained more in detail.

Our reply:

Following the Referee's recommendation the manuscript has been revised considerably. For the relevant revisions related to this points we remit the Reviewer to our response to points: **R#2c)** and **R#2d)** of the Reviewer #2.

R#3-2, I would suggest authors to give some explanations for Fig. 1 where hysteresis loop are observed at 2 K for both (H//c and H perpendicular to c) while T_C is only 1 K for this sample. This will help non-experts to follow the idea.

Our reply:

To clarify this issue we have supplemented the manuscript by two new paragraphs (p. 3, section "Magnetization as a function of the magnetic field", which read:

"The circles in Fig. 1 represent the experimental data $M(T,H)$ per unit sample area A for the A and B $Ga_{1-x}Mn_xN$ films at two orientations of the magnetic field H with respect to the c -axis. No hysteretic behaviour is found down to the experimental noise level [$\sigma_M/A(H \approx 0) \leq 1 \times 10^{-6} \text{ emu/cm}^2$, where $A \approx 0.25 \text{ cm}^2$] at any investigated temperature (2, 5, 10 K – presented in Fig. 1 -- and at 50 and 300 K, not shown). This points to the absence of a ferromagnetic phase that might originate from Mn-rich aggregates. This conclusion is further supported by results of HRXRD and HRTEM studies on these samples presented in Methods, and which -- within the ultimate limits specific to these techniques -- rule out the presence of any Mn-rich phases either in the form of nanometric precipitates or of regions of spinodal decomposition¹².

Actually, both layers exhibit magnetism characterized by $M(T, 0) = 0$, $M(T; H \rightarrow \infty)$ tends to saturate, and $\chi(T) = dM(T, H \rightarrow 0)/dH$ increases with lowering temperature according to $\chi(T) / (T - T_0)$. This behaviour demonstrates that the studied samples at $T \geq 2 \text{ K}$ are in the paramagnetic phase, though ferromagnetic coupling between neighbouring Mn spins starts to be visible and leads to $T_0 > 0$. At the same time, a sizable uniaxial magnetic anisotropy is observed, $M(T, H \perp c) > M(T, H \parallel c)$, which is specific for Mn^{3+} ions in wz GaN (refs 7 and 8).

R#3-3, Some parameters such as $B^0_2 = 3.2 \text{ mV}$ (on page 2), and spin-orbit parameters $\lambda_{\{TT\}}$ etc. should be introduced and explained in detail.

Our reply:

This recommendation of the Reviewer is followed by supplementing the manuscript by a new subsection in Methods, "Theoretical model and relevant material parameters" (pp 9 and 10).

R#3-4, It seems to me that the theoretical fit to experimental data in Fig. 3c for H//c is not good, especially for $H > 15 \text{ KOe}$. More explanations are welcomed. Actually, the fit to 5 K data in Fig. 4 is not good as well.

Our reply:

Agreeing with the Reviewer suggestion we have added the relevant statement in the section Discussion (p. 6)

“The level of achieved agreement indicates that the PEME dominates in the studied $Ga_{1-x}Mn_xN$ films. At the same time, it is clear that in some cases there is only qualitative agreement between experimental results and theoretical expectations (see, e.g., high field data in Fig. 3c, at 5 K in Fig. 4, and in high electric fields in Fig. 5). On the experimental side, as already noted, surface roughness detected by AFM may lead to lateral non-uniformities of the applied electric field, which can affect the behaviour of the magneto-electric effect. It is also possible that the observed discrepancies indicate that the piezoelectric stretching not only changes the magnitude of trigonal distortion but also alters e.g., the Jahn-Teller effect or spin-spin coupling. Finally, the discrepancies may herald an onset of a contribution from a not yet identified phenomenon beyond PEME.”

R#3-5, Typo: On page 1, the first paragraph, " First, the magnetoelectric effect generated by piezoelectricity can exist is a single compound". Here "is" should be "in".

Our reply:

Now the mentioned sentence reads:

“The magnetoelectric effect generated by piezoelectricity can exist in a homogeneous crystalline compound...”

We would like to thank once more the Reviewers for the constructive comments and recommendations, and we hope that the new version of the manuscript meets the requirements for publication in Nature Communications.

Reviewers' Comments:

Reviewer #1 (Remarks to the Author):

The authors have improved the manuscript greatly based on the reviewers comments. In my opinion, it is now very clear and should be published as is.

Reviewer #2 (Remarks to the Author):

[The Reviewer believes the manuscript is now suitable for publication with no further comments for the authors.]

Reviewer #3 (Remarks to the Author):

[The reviewer believes the paper is ready to be published, with no further comments for the authors]